# Biotin methyl ester enhances cargo release in RUSH system and enables rapid biotinylation with TurboID
Taisei Uehara [1,4], Arata Takiguchi[1,4], Takuro Tojima [2], Akihiko Nakano [2,3], Shion Kagawa[1], Tatsuo Nehira [1], Takunori Satoh [1,5] & Akiko K. Satoh [1,5]

Biotin-streptavidin technology is a key tool in biotechnology. The engineering of streptavidin or streptavidin-like molecules and organic synthesis of biotin derivatives offer a variety of applications to meet the needs of researchers. Here, we report that owing to its masked charged carboxyl group, biotin methyl ester (BME) improves cell penetration and allows the retention using selective hooks (RUSH) system to initiate cargo release promptly and uniformly in all cells. In addition, because of the rapid hydrolysis of BME to biotin inside cells, BME enabled faster biotinylation than biotin, by the promiscuous biotin ligase TurboID. These properties allow BME to act as a biotin prodrug, enhance the RUSH system and TurboID, and highlight the potential of biotin-streptavidin technology for numerous applications in living cells. Additionally, BME may be a good biotin supplement for patients with multivitamin-responsive inherited metabolic disorders.

Biotin is a vital vitamin essential for life[1]. It serves as a coenzyme for several carboxylases involved in critical metabolic reactions. As a result, organisms that are incapable of biotin de novo must acquire biotin from the external environment[2]. Beyond its metabolic role, biotin has garnered significant attention in biochemistry and cell biology because of its exceptionally strong affinity for streptavidin. This biotin-streptavidin interaction has become a cornerstone of biotechnology[3,4], enabling the development of numerous in vitro and in vivo techniques for isolating biotinylated proteins and releasing streptavidin-binding peptides (SBP) from streptavidin[5–7].

One of the methods used to exploit the ability of biotin to release SBP from streptavidin in living cells is the retention using selective hooks (RUSH) system, which allows the synchronous release of cargo from organelles[8]. The RUSH system has been widely used to track secretory cargo in live cell imaging[9–13]. However, we recently reported an unexpected variation in cellular response to biotin[14]. HeLa cells with high cargo and hook expression levels, after transient transfection with the original RUSH constructs, showed longer delays in inducing cargo export from the endoplasmic reticulum (ER) after biotin administration. Because cargo release can start only after biotin excludes SBP from streptavidin in the ER by occupying most of the binding sites, high expression levels of the streptavidin hook would require more biotin to release the cargo. This observation led us to wonder whether the lack of temporal control in the RUSH system

was caused by an insufficient supply of biotin to the cells. Although biotin uptake by mammalian cells is primarily mediated by the sodium multivitamin transporter SMVT (encoded by *SLC5A6* in humans)[15–18], the activity of this transporter may not always be sufficiently high to allow for prompt cargo release immediately after biotin addition to the medium. Transporter-independent plasma membrane permeation of biotin is expected to be slow because biotin has a carboxyl group with an acid dissociation constant of 4.7, and therefore carries a negative charge at physiological pH (Fig. 1a). One strategy for delivering drugs with carboxyl groups or other acid residues into cells is to use esterified drugs as prodrugs, which can easily enter cells and be hydrolyzed by carboxylesterases (CESs)[19–21]. Therefore, we aimed to assess the use of biotin methyl ester (BME) as a biotin prodrug. Even if BME is difficult to hydrolyze to biotin within cells, it is expected to improve the temporal control of the RUSH system, because the carboxyl group of biotin itself is not involved in its binding to streptavidin[22]. Of note, it has already been demonstrated that BME can release cargo in the RUSH system, although whether BME is more effective than biotin in cargo release has not been addressed[23,24].

Proximity biotinylation is another widely used biotin-dependent method for the identification of partner protein[25]. BioID (proximity-dependent biotin identification), a mutant of the *Escherichia coli* biotin ligase BirA, its derivatives with higher activity (e.g., TurboID and AirID) and

[1]Graduate School of Integrated Sciences for Life, Hiroshima University, Higashi-Hiroshima, Hiroshima, Japan. [2]Live Cell Super-Resolution Imaging Research Team, RIKEN Center for Advanced Photonics, Wako, Saitama, Japan. [3]Institute of Integrated Research, Institute of Science Tokyo, Chiyoda-ku, Tokyo, Japan. [4]These authors contributed equally: Taisei Uehara, Arata Takiguchi. [5]These authors jointly supervised this work. Takunori Satoh, Akiko K. Satoh.
✉e-mail: tsatoh3@hiroshima-u.ac.jp; aksatoh@hiroshima-u.ac.jp

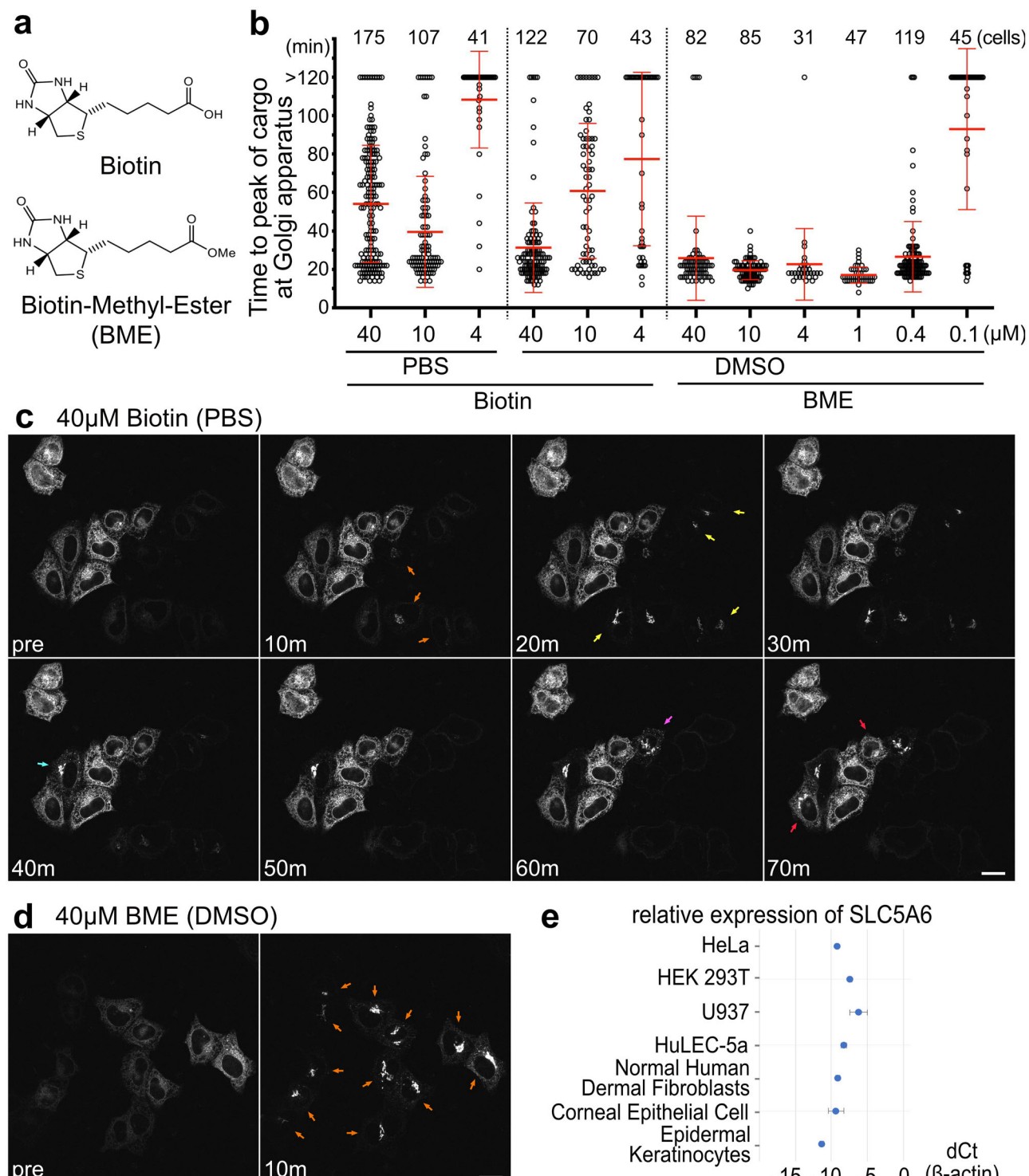

**Fig. 1 | Synchronous release of cargos in the RUSH system by BME in HeLa cells.**
**a** Structure of biotin and BME. **b** Plot of time to peak cargo SBP::mCherry::GPI in the Golgi apparatus after administration of biotin or BME in the RUSH system in HeLa cells. The numbers of cells observed under each condition are shown at the top of each plot. Biotin solutions concentrated in PBS or DMSO were used to obtain the final concentrations, which are indicated at the bottom of each plot. Cells that did not reach maximum intensity in the Golgi area were counted as >120 min. Error bars are mean ± SD. Results of Dunn's multiple comparisons test for (Biotin 40 µM/DMSO vs. Biotin 4 µM/DMSO), BME 40 µM/DMSO vs. BME 0.1 µM/DMSO), (Biotin 10 µM/DMSO vs. BME 10 µM/DMSO), (Biotin 4 µM/DMSO vs. BME 4 µM/DMSO) are $p < 0.0001$, and those for (Biotin 40 µM/DMSO vs. Biotin 10µM/DMSO), (BME 40 µM/DMSO vs. BME 1 µM/DMSO) are $p < 0.001$, and those for (Biotin 40 µM/DMSO vs. BME 40 µM/DMSO), (BME 40 µM/DMSO vs. BME

10 µM/DMSO), (BME 40 µM/DMSO vs. BME 4 µM/DMSO), (BME 40 µM/DMSO vs. BME 0.4 µM/DMSO) are not significant. **c** SBP::EGFP::GPI localizations before (upper left) and at 10, 20, 30, 40, 50, 60, and 70 min after addition of 40 µM biotin in the RUSH system in HeLa cells. Arrows indicate cells in which the SBP::EGFP::GPI cargo started to accumulate in the Golgi apparatus at that time point.
**d** SBP::EGFP::GPI localizations before (left) and at 10 min after addition of 40 µM of BME in the RUSH system in HeLa cells. Arrows indicate cells in which the S SBP::EGFP::GPI cargo began to accumulate in the Golgi apparatus. **e** mRNA expression of *SLC5A6* in various cell lines. *ACTB* was used as an endogenous control for the total mRNA levels. Scale bars: 20 µm (**c**, **d**). BME, biotin methyl ester; DMSO, dimethyl sulfoxide; PBS, phosphate-buffered saline; RUSH, retention using selective hooks.

its ortholog in *Aquifex aeolicus* with similar mutation (e.g., UltraID) exhibit promiscuous activity and release highly reactive and short-lived biotinyl-5'-AMP[26–30]. Released biotinyl-5'-AMP modifies proximal proteins within approximately 10 nm, which can be easily isolated using streptavidin beads and identified via mass spectrometry[26–28,31]. Biotinylation in living cells is initiated by the addition of biotin to the medium, which is supposed to be rapidly taken up by the cells and become the substrate for these enzymes to produce biotinyl-5'-AMP. However, slow uptake of biotin into cells may limit the biotinylation rate. Unlike streptavidin-binding in the RUSH system, BME is unlikely to be used directly as a substrate for BioID-derived enzymes. In this case, rapid hydrolysis of BME to biotin is essential. We investigated whether BME improves the rate of TurboID-mediated biotinylation in living cells.

## Results

### Synchronous release of cargo in the RUSH system by BME

Membrane and secretory proteins synthesized in the ER are transported to the *cis*-side of the Golgi apparatus, undergo various modifications, and exit from the *trans*-side to their respective destinations. Although tracking secretory cargoes using high-resolution microscopy can help us understand the molecular mechanisms of cargo transport, it still requires either pulse labeling or synchronous release of cargo proteins to ensure clear visualization[32–35]. The RUSH system is the most widely used method for synchronous cargo release with streptavidin, SBP, and biotin[8].

In the RUSH system, streptavidin, coupled with an organelle-specific retention signal, effectively sequesters SBP-tagged cargo proteins. The introduction of biotin into the medium induced a synchronized release of SBP-tagged cargo proteins from the organelle via competitive binding with high affinity to the SBP-binding sites on streptavidin. Cargo proteins fused to fluorescent proteins or tags allow the real-time imaging of their journey from sequestered organelles to their destination. This is an ingenious way of exploiting the difference in affinity between biotin and SBP for streptavidin, and we are one of the groups that have benefited from the RUSH system[36], although we have also found that not all cells respond immediately to biotin at a final concentration of 40 μM, the concentration recommended in the original RUSH paper (Fig. 1b, c and Supplementary Video 1)[14]. The plot of maximum fluorescence intensity over time in the Golgi area shows strong variability at 10 or 40 μM biotin and no movement at 4 μM biotin in most cells (Fig. 1b). In contrast, when BME was added to the medium instead of biotin at a final concentration of 40 μM, most of cells responded immediately (Fig. 1d). The plot shows the well synchronized cargo accumulation in the Golgi region of all cells at around 20 min even at a final concentration of 1 μM (Fig. 1b). Dimethyl sulfoxide (DMSO) was used as the solvent for BME stock solutions because it does not dissolve well in phosphate buffered saline (PBS), which is normally used as a solvent for biotin. As the difference in solvent might affect the responsiveness of cells to biotin or BME, we also examined cargo release using a biotin stock dissolved in DMSO (Fig. 1b). The results showed that 40 μM biotin with 0.04% DMSO in medium exhibited a better cellular response than 40 μM biotin in medium without DMSO, but the effect was still much weaker than that for 1 μM BME with 0.04% DMSO in medium (Fig. 1b). In fact, 0.4 μM BME, a 100-fold lower concentration, has a similar effect to 40 μM biotin with 0.04% DMSO in medium. These results indicate that the poor uptake rate of biotin is the reason for the variability in the cellular response to biotin, and that BME significantly improved the cellular response in the RUSH system.

### *SLC5A6* expression in HeLa cells compared to other human culture cells

Biotin uptake is primarily mediated by the sodium multivitamin transporter encoded by *SLC5A6* in humans[15,18]. The limitations in biotin uptake may be specific to HeLa cells. Therefore, we investigated the expression of *SLC5A6* in other human cell lines. We quantified the mRNA expression of *SLC5A6* in normal endothelial, fibroblast, epithelial, and epidermal cells as well as in the common tumor cell lines HeLa, HEK293T, and U937. The relative expression of *SLC5A6* in HeLa cells was approximately 1/4 and 1/8 of that in

HEK293T and U937 cells, respectively, comparable to that in epithelial, epidermal, and endothelial cells (Fig. 1e). Thus, the limited availability of biotin in HeLa cells cannot be explained by the low cell-line-specific expression of *SLC5A6*.

We also examined the effects of BME on MDCK and HEK293T cells. We observed that not all of the cells respond immediately to biotin at a final concentration of 40 μM (Supplementary Fig. S1a, c and Supplementary Video 2, 3). In contrast, most MDCK and HEK293T cells respond immediately after 40 μM BME administration (Supplementary Fig. S1b, d and Supplementary Video 2, 3). Thus, the enhanced cargo release by BME in the RUSH system is not limited to HeLa cells.

Although the BME molecule is expected to have increased membrane permeability compared to biotin, its uptake may still be mediated by SMVT. Thus, we investigated whether pantothenic acid, which is known to competitively inhibits biotin transport mediated by SMVT[37], would also inhibit the cargo release induced by BME. As expected, simultaneously adding 1 mM pantothenic acid severely inhibited cargo release induced by 40 μM biotin (Supplementary Fig. S2b and Supplementary Video 4). In contrast, 40 μM BME effectively released cargo in all cells, even in the presence of 1 mM pantothenic acid (Supplementary Fig. S2d and Supplementary Video 5). These results suggest that BME entry into cells does not depend on SMVT and that BME penetrates directly into cells due to its hydrophobicity.

### Enhancement of TurboID biotinylation in living cells by BME

The considerable improvement in cargo release in the RUSH system when BME was used instead of biotin suggests that the uptake of biotin into cells may be the rate-limiting step in other methods that use biotindirectlyin living cells. Therefore, we investigated whether proximity biotinylation by BirA-derived enzymes could also be improved by BME instead of biotin. The high activity of TurboID, which can biotinylate proteins within 10 min, necessitates rapid availability of biotin. ER-localized TurboID with green fluorescent protein, clover (Cv), and the KDEL sequence (Cv::TurboID::KDEL) were chosen to evaluate biotin ligase activity. We incubated the cells with 1 to 400 μM of biotin or BME for 10 min, and biotinylated proteins were detected with Alexa Fluor 568-conjugated streptavidin (Fig. 2a). Much stronger signals were detected with BME than with biotin at all tested concentrations; however, the difference was more pronounced at lower concentrations. The Alexa Fluor 568 signal was quantified, and at final concentrations of 1, 5, 10, and 50 μM, 3.53, 17.25, 5.68, and 12.1 times more signals, respectively, were detected in BME-treated cells than in biotin-treated cells at the same concentration (Fig. 2b).

Biotinylated proteins were detected by immunoblotting (Fig. 2c, top). The major band marked with an arrow is Cv::TurboID::KDEL, which resulted from self-biotinylation (Fig. 2c, arrow). The signal of this band was much stronger in BME-treated cells than in biotin-treated cells at the same concentrations, and this difference was more pronounced at lower concentrations. We measured and plotted the signal intensity of the Cv::TurboID::KDEL band in the region surrounded by the red rectangle (Fig. 2d and Supplementary Fig. 3a). The results showed that 5.86, 5.61, and 4.26 times more biotinylation occurred in BME-treated cells than in biotin-treated cells at 1, 5, and 10 μM, respectively (Fig. 2d). We also measured and plotted the signal intensity on the immunoblotted membrane, which represented the non-auto-biotinylation of proteins in the region surrounded by a blue rectangle (Fig. 2e, Supplementary Fig. 3a). The results indicated that 5.69, 4.55, and 2.88 times more biotinylation occurred in BME-treated cells than in the biotin-treated cells at 1, 5, and 10 μM (Fig. 2e). The immunoblotting results were comparable to those obtained by cell staining followed by confocal microscopy. These results indicate that BME enhances the biotinylation efficiency of TurboID in living cells.

### BME is not a substrate for TurboID in vitro

The carboxyl group of biotin, the reactive group for generating biotinyl-5'-AMP before attachment to lysin residues of nearby proteins, is protected in BME. Therefore, BME itself is not supposed to be a substrate for TurboID, although strong biotinylation was observed in living cells (Fig. 2a–e). To

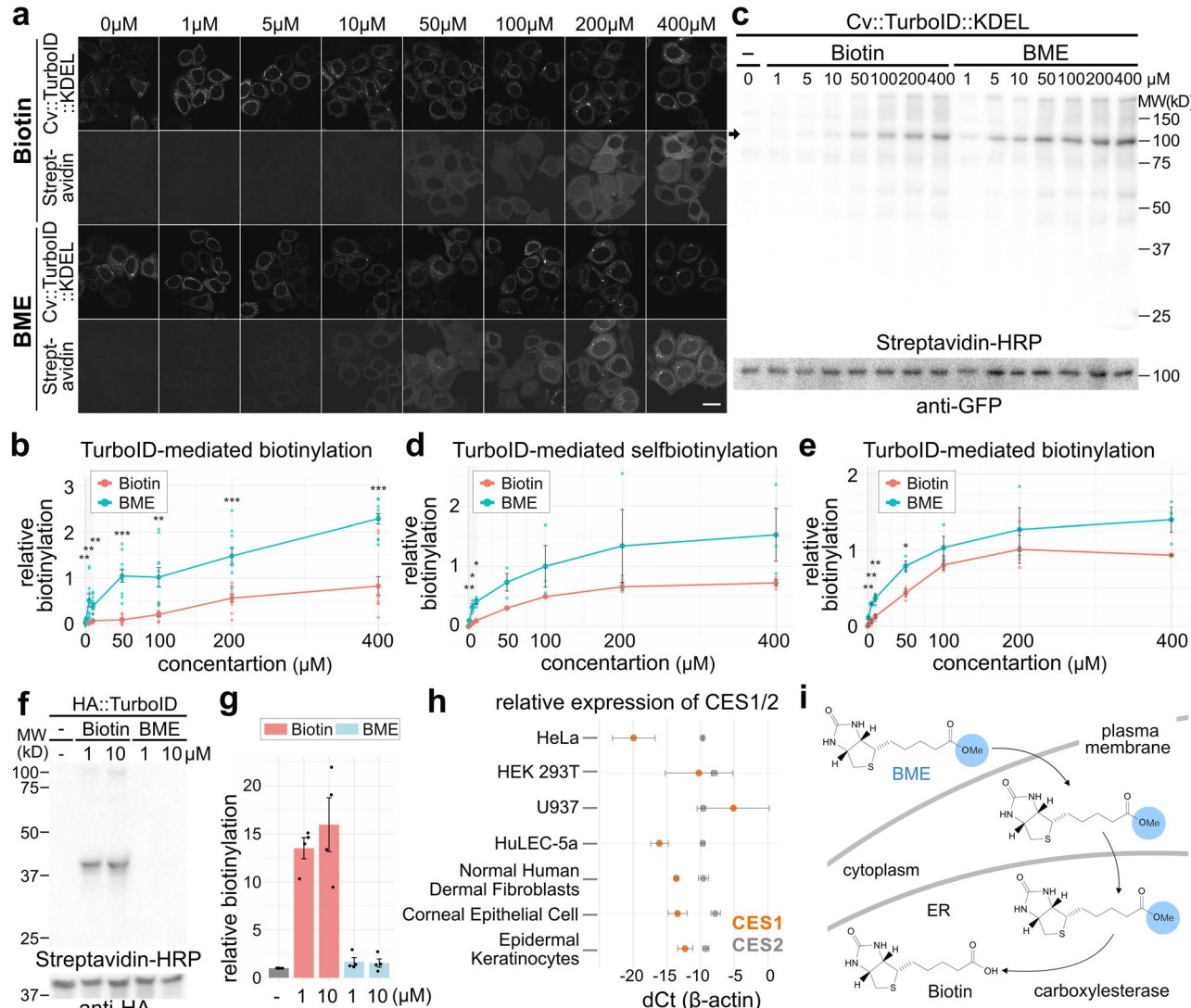

**Fig. 2 | BME enhances TurboID biotinylation in living cells. a** Detection of biotinylated proteins by confocal microscopy. HeLa cells expressing Cv::TurboID::KDEL were fixed after 10 min incubation with biotin or BME at 0, 1, 5, 10, 50, 100, 200, and 400 μM, and biotinylated proteins were detected using Alexa Fluor 568-conjugated streptavidin. Top panel: fluorescence signal of Cv::TurboID ::KDEL and bottom panel: fluorescence signal of Alexa Fluor 568 for biotinylated proteins. **b** Relative amount of biotinylation detected by confocal microscopy. Significance was determined using a two-tailed paired Student's $t$ test. Exact $P$ values are 0.0013 (1 μM), 0.0088 (5 μM), 0.0013 (10 μM), 0.00004 (50 μM), 0.0054 (100 μM), 0.0008 (200 μM), and 0.0005 (400 μM). **c** Detection of biotinylated proteins by immunoblotting. Extracts from HeLa cells expressing Cv::TurboID::KDEL were prepared after 10 min incubation with biotin or BME at 0, 1, 5, 10, 50, 100, 200, and 400 μM. Streptavidin-HRP or anti-GFP antibodies were used to detect biotinylated proteins (top) or Cv::TurboID::KDEL (bottom), respectively. The arrow indicates the band for Cv::TurboID::KDEL. **d** Relative amount of Cv::TurboID::KDEL self-biotinylation. The intensities of Cv::TurboID::KDEL (red rectangles in Fig. S3a) were measured in three independent experiments. Significance was determined using a two-tailed paired Student's $t$ test. Exact $P$ values are 0.0062 (1 μM), 0.0459 (5 μM),

0.0232 (10 μM), 0.0883 (50 μM), 0.2880 (100 μM), 0.4049 (200 μM), and 0.1729 (400 μM). **e** Relative amount of biotinylation. The intensity of the large areas on the immunoblot membrane (blue rectangles in Fig. S3a) were measured in three independent experiments. Significance was determined using a two-tailed paired Student's $t$ test. Exact $P$ values are 0.0079 (1 μM), 0.0032 (5 μM), 0.0100 (10 μM), 0.0339 (50 μM), 0.3238 (100 μM), 0.5935 (200 μM), and 0.10749 (400 μM). **f** Detection of biotinylated proteins via immunoblotting. HA::TurboID::KDEL synthesized by in vitro translation was incubated with biotin or BME at 0, 1, and 10 μM. Streptavidin-HRP or anti-HA antibodies were used to detect biotinylated proteins (top) or HA::TurboID (bottom), respectively. **g** Relative amount of HA::TurboID self-biotinylation. HA::TurboID intensity measured in four independent experiments. **h** *CES1* (orange) and *CES2* (gray) mRNA expression in various cell lines. *ACTB* was used as an endogenous control for total mRNA levels. **i** Model of BME activity as a prodrug. BME enters the cells and is hydrolyzed to biotin. Error bars are presented as mean ± SE in (**b**, **d**, **e**, **g**) and mean ± SD in **h**. Significance according to the two-tailed paired Student's $t$ test: *$p < 0.05$, **$p < 0.0$, ***$p < 0.001$. Scale bars: 20 μm. BME, biotin methyl ester; HRP, horseradish peroxidase.

investigate whether TurboID can directly use BME as its substrate, an in vitro biotinylation assay was performed using HA::TurboID translated into the PURE system, which is a chemically defined cell-free in vitro translation system[38]. The results showed that HA::TurboID biotinylated proteins with biotin, but not with BME (Fig. 2f, g). Strong biotinylation within 10 min in living cells and no biotinylation in vitro by TurboID with BME suggested that BME entered cells and was hydrolyzed to biotin within 10 min.

Compounds esterified between small alcohol groups and large acyl groups are typically hydrolyzed by carboxylesterase 1 (CES1), whereas compounds esterified between large alcohol groups and small acyl groups are typically hydrolyzed by CES2[21,39]. Because BME is an esterified compound between small alcohol and large acyl groups, it is more likely to be hydrolyzed by CES1. Since the enhancement of TurboID biotinylation efficiency by BME in living cells depends on the expression of *CES1* and

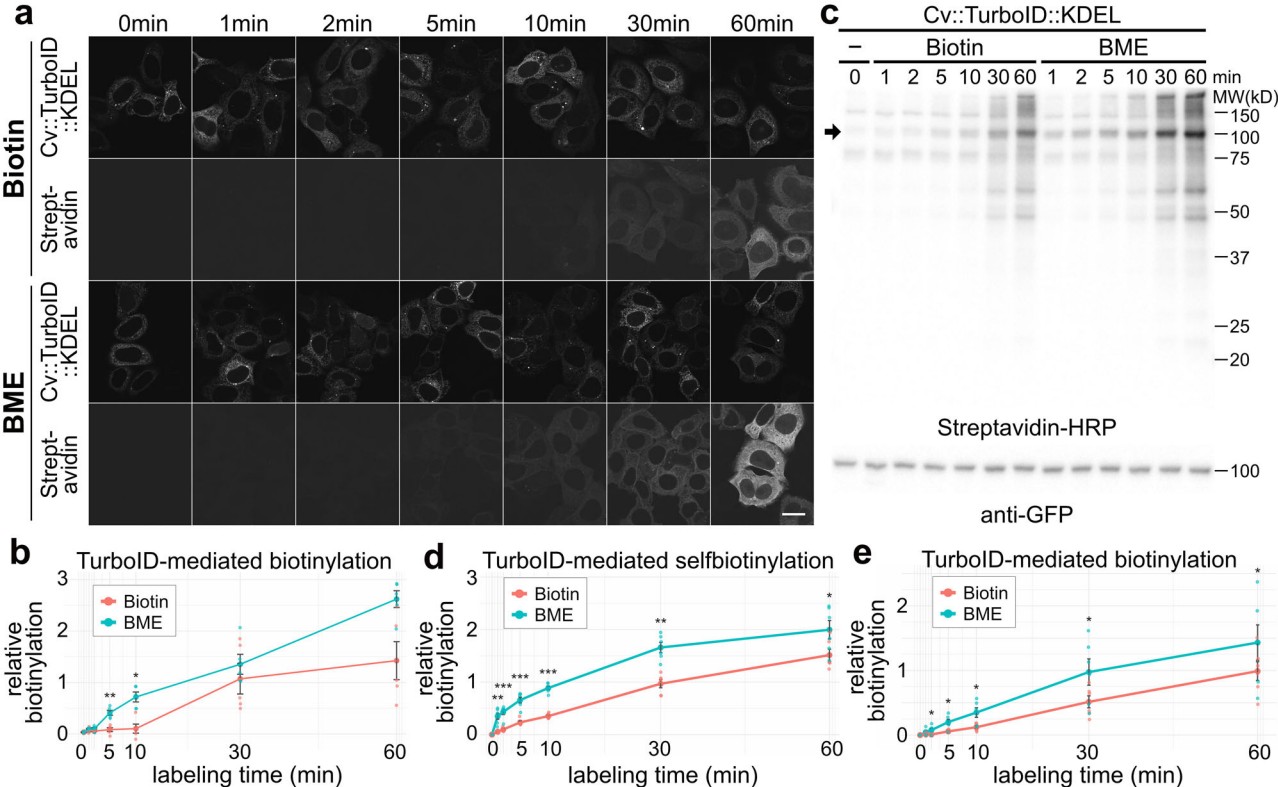

**Fig. 3 | BME enables rapid biotinylation with TurboID in HeLa cells. a** Detection of biotinylated proteins by confocal microscopy. HeLa cells expressing Cv::TurboID::KDEL were fixed after 0, 1, 2, 5, 10, 30, and 60 min incubation with biotin or BME at 50 μM, and biotinylated proteins were detected using Alexa Fluor 568-conjugated streptavidin. Top panel: fluorescence signal of Cv::TurboID::KDEL and bottom panel: fluorescence signal of Alexa Fluor 568 for biotinylated proteins. **b** Relative amount of biotinylation detected by confocal microscopy. Significance was determined using a two-tailed paired Student's *t* test. Exact *P* values are 0.1411 (1 min), 0.1000 (2 min), 0.0014 (5 min), 0.0217 (10 min), 0.5045 (30 min), and 0.0652 (60 min). **c** Detection of biotinylated proteins by immunoblotting. Extracts from HeLa cells expressing Cv::TurboID::KDEL were prepared after 0, 1, 2, 5, 10, 30, and 60 min incubation with biotin or BME at 50 μM. Streptavidin-HRP or anti-GFP antibodies were used to detect biotinylated proteins (top) or Cv::TurboID::KDEL

(bottom), respectively. The arrow indicates the band for Cv::TurboID::KDEL. **d** Relative amount of Cv::TurboID::KDEL self-biotinylation. The intensities of Cv::TurboID::KDEL (red rectangles in Fig. S3b) were measured in six independent experiments. Significance was determined using a two-tailed paired Student's *t* test. Exact *P* values are 0.0030 (1 min), 0.0008 (2 min), 0.0004 (5 min), 0.000007 (10 min), 0.0016 (30 min), and 0.0189 (60 min). **e** Relative amount of biotinylation. The intensity of the large areas on the immunoblot membrane (blue rectangles in Fig. S3b) were measured in six independent experiments. Significance was determined using a two-tailed paired Student's *t* test. Exact *P* values are 0.0636 (1 min), 0.0267 (2 min), 0.0123 (5 min), 0.0146 (10 min), 0.0310 (30 min), and 0.0287 (60 min). Error bars are presented in (**b**, **d**, **e**) as mean ± SE. Significance was determined using a two-tailed paired Student's *t* test: \**p* < 0.05, \*\**p* < 0.01, and \*\*\**p* < 0.001. Scale bars: 20 μm. BME, biotin methyl ester.

*CES2*, we compared the expression of these genes in the common tumor cell lines (HeLa, HEK293T, and U937) and in normal endothelial, fibroblast, epithelial, and epidermal cells. The expression of *CES2* was comparable among the cell types (Fig. 2h, gray dots). However, the expression of *CES1* in HeLa cells was much lower than that in other cells (Fig. 2h, orange dots). Thus, rapid BME hydrolysis is not specific to HeLa cells; rather, it is common in most cells. In short, because BME easily enters cells and is quickly hydrolyzed to biotin, it functions more efficiently than biotin as a TurboID substrate (Fig. 2i).

### BME enables rapid TurboID biotinylation in living cells

The high activity of TurboID has motivated researchers to understand the rapid cellular responses after drug administration and rapid changes in cellular conditions. Therefore, we assessed how BME improved the time resolution of TurboID-mediated biotinylation. As judged by the synchronicity of the cargo start in the RUSH system (Fig. 1), the cell permeability of BME was high; however, the conversion of BME to biotin could be the rate-limiting step for TurboID. We compared the efficiency of biotinylation at various incubation times after BME administration, particularly at early time points (< 10 min).

We incubated the cells with 50 μM of biotin or BME for 1 to 60 min, and biotinylated proteins were detected with Alexa Fluor 568-conjugated

streptavidin (Fig. 3a). Stronger fluorescence signals were detected with BME than with biotin at all the time points. The stronger biotinylation by BME than by biotin was evident at 5 and 10 min: 4.90 and 6.94 times more biotinylation occurred in BME-treated cells compared to biotin-treated cells (Fig. 3b).

The biotinylated proteins were detected by immunoblotting (Fig. 3c, top panel). Higher biotinylation activity was observed with BME than with biotin at all the time points. Enhanced biotinylation by BME compared to that by biotin was much more pronounced at short incubation times. We measured and plotted the signal intensity of the Cv::TurboID::KDEL band in the region surrounded by the red rectangle (Fig. 3d and Supplementary Fig. 3b). The results showed that 6.43, 4.41, 2.83, and 2.52 times more biotinylation occurred in BME-treated cells than in biotin-treated cells at 1, 2, 5, and 10 min, respectively (Fig. 3d). We also measured and plotted the signal intensity on the immunoblotted membrane, which represented the non-auto-biotinylation of proteins in the region surrounded by a blue rectangle (Fig. 3e, Supplementary Fig. 3b). The results showed 4.91, 8.65, 3.51, and 2.87 times more biotinylation occurred in BME-treated cells compared to that in biotin-treated cells at 1, 2, 5, and 10 min, respectively (Fig. 3e). These results indicate that BME is rapidly hydrolyzed to biotin inside cells, and that BME enables rapid biotinylation by TurboID in living cells.

## BME enables rapid TurboID biotinylation in MDCK and HEK293T cells

We examined whether BME mediates rapid biotinylation in other cell types. We compared the efficiency of biotinylation at various time points after administering biotin or BME to MDCK and HEK293T cells. In MDCK cells, BME exhibited a very high biotinylation efficiency at all time points (Supplementary Fig. S4a). The plot of Cv::TurboID::KDEL signal intensity (red rectangles in Supplementary Fig. S5a) indicated that 3.07, 4.04, 6.13, 6.57, 4.96, and 5.22 fold greater levels of biotinylation occurred in BME-treated cells compared to that in biotin-treated cells at 1, 2, 5, 10, 30, and 60 min, respectively (Supplementary Fig. S4b). Conversely, in HEK293T cells, the biotinylation efficiency of BME depended on the incubation time. A short incubation time (1 or 2 min) enhanced biotinylation with BME; however, a longer incubation time resulted in stronger biotinylation with biotin in HEK293T cells (Supplementary Fig. S4c). The plot of Cv::TurboID::KDEL signal intensity (red rectangles in Supplementary Fig. S5b) indicated that BME-treated cells underwent 2.94, 1.08, 0.67, 0.86, 0.71, and 0.75 fold more biotinylation compared to that of biotin-treated cells at 1, 2, 5, 10, 30, and 60 min, respectively (Supplementary Fig. S4d).

HEK293T cells exhibited an enhanced cargo-releasing effect of BME in the RUSH system (Supplementary Fig. S1c, d), and BME must enter the cells much faster than biotin. Although CES2 and CES1 expression was comparable between HeLa and HEK293T cells in our experiments (Fig. 2h), it has been demonstrated that HEK293T cells possess low carboxylesterase activity[40]. Thus, the weaker effect of BME on biotinylation enhancement could be explained by its low carboxylesterase activity. Therefore, we expressed CES2 in HEK293T cells and investigated the efficiency of biotinylation of BME and biotin. We observed better biotinylation efficiency in HEK293T cells expressing CES2 (Supplementary Fig. S4e). Signal intensity plots of the Cv::TurboID::KDEL band (Supplementary Fig. S5c) revealed that BME-treated cells exhibited 1.21, 1.47, 1.39, 1.27, 1.13, and 0.80 fold higher biotinylation rates than those of biotin-treated cells at 1, 2, 5, 10, 30, and 60 min, respectively (Supplementary Fig. S4f). These results indicate that efficient biotinylation by BME depends on carboxylesterase activity and that BME always exhibits higher biotinylation efficiency at shorter incubation times. Thus, BME is suitable for rapid biotinylation.

## BME effects on TurboID biotinylation activity in cytosol or nucleus

Esterified prodrugs are typically hydrolyzed by CES1 and CES2. These enzymes are localized in the ER via the HXEL sequence binding to KDEL receptors[19-21], suggesting that BME is hydrolyzed within the ER. Consequently, the biotin concentration may be higher in the ER lumen than in the cytosol or other organelles. If this is the case, the enhancing effect of BME on the rapid biotinylation by TurboID may depend on its localization. To investigate this possibility, we examined the effects of BME on rapid biotinylation by TurboID, which is localized in the cytosol and nucleus.

We constructed ARF1::TurboID::Cv as the cytosolic-localizing TurboID and Histone H2B (H2B)::TurboID::Cv as the nuclear-localizing TurboID. As expected, ARF1::TurboID::Cv was localized mainly in the Golgi apparatus but also slightly in the cytoplasm (Fig. 4a, green). H2B::TurboID::Cv was exclusively localized in the nucleus (Fig. 4b, green). Next, we investigated the distribution of biotinylated proteins by ARF1::TurboID::Cv and H2B::TurboID::Cv. Cells expressing ARF1::TurboID::Cv or H2B::TurboID::Cv were incubated with 50 µM BME for 10 min, and biotinylated proteins were detected with Alexa Fluor 568-conjugated streptavidin. Signals were found in both the Golgi apparatus and cytoplasm of cells expressing ARF1::TurboID::Cv (Fig. 4a, magenta). In contrast, in cells expressing H2B::TurboID::Cv, signals were found only in the nucleus (Fig. 4b, magenta).

Proteins biotinylated by ARF1::TurboID::Cv and H2B::TurboID::Cv were detected using immunoblotting (Fig. 4c and d). Higher biotinylation activity of both TurboIDs was observed with BME than with biotin at all time points. Enhanced biotinylation by BME compared to biotin was more pronounced at shorter incubation times in H2B::TurboID::Cv. We

measured and plotted the signal intensities of the ARF1::TurboID::Cv and H2B::TurboID::Cv bands in the region surrounded by the red rectangle (Fig. 4e, f, and Supplementary Fig. 5d, e). The results indicated that 2.66, 2.51, 2.50, and 1.92 times more biotinylation for ARF1::TurboID::Cv and 4.26, 2.33, 1.90, and 2.01 times more biotinylation for H2B::TurboID::Cv occurred in BME-treated cells than in biotin-treated cells at 1, 2, 5, and 10 min, respectively (Fig. 4e, f). We also measured and plotted the signal intensity on the immunoblotted membrane, which represents the non-autobiotinylation of proteins in the region surrounded by a blue rectangle (Fig. 4g, h, and Supplementary Fig. 5d, e). The results indicated that 1.88, 1.70, 2.68, and 2.19 times more biotinylation for ARF1::TurboID::Cv and 2.17, 1.70, and 2.34 times more biotinylation for H2B::TurboID::Cv occurred in BME-treated cells than in biotin-treated cells at 2, 5, and 10 min, respectively (Fig. 4g, h). These results indicate that biotin hydrolyzed from BME is rapidly available to both cytosolic and nuclear TurboIDs. These results indicated that BME is a general tool for rapid biotinylation by TurboID in living cells.

## Discussion

We demonstrated an improvement in the RUSH system and TurboID biotinylation by using BME instead of biotin. Our results collectively indicate that the low cellular uptake rate of biotin limits the release of SBP-tagged cargo from streptavidin hooks in the RUSH system as well as the biotinylation activity of TurboID. BME easily penetrates the cells because of its protected carboxyl group and is rapidly hydrolyzed to biotin. Therefore, BME acts as a prodrug or pro-substrate for biotin. Figure 5 illustrates the principles of the RUSH system and TurboID, as well as the mechanism by which BME improves these systems.

The ability to initiate transport in all cells simultaneously is particularly useful in experiments where cells are fixed and observed at a specific time after reagent application. When transport is initiated by BME, all cells begin transport promptly after BME application; thus, the time after BME administration represents the time after the start of transport. Whereas with biotin administration, the timing of transport initiation in different fixed cells is much more varied in the RUSH system. Even in live imaging, the simultaneous release of cargo in all cells makes observation easier, as researchers can decide which cells to observe before adding the reagent, without having to look for cells with cargo movement. In addition, BME allows the quantitative analysis of the exit rate of each type of cargo from the ERES, or the time the cargo reaches the cis-Golgi, etc. Thus, the BME makes the existing RUSH system a more powerful method.

Proximity biotinylation is widely used to identify partner proteins[25]. BME has the potential to increase the biotinylation rate of all BirA-derived enzymes, including TurboID, UltraID, and AirID, owing to its ability to rapidly provide substrates. The possible BME-hydrolyzing enzymes, CES1 and CES2, are localized in the ER, resulting in a higher concentration of biotin in the ER lumen than in the cytosol or other organelles. Nevertheless, our comparison of the effects of BME with those of biotin on the biotinylation activity of TurboID localized in the ER, cytosol, and nucleus indicate that, wherever TurboID is localized, BME strongly enhances biotinylation rates. The ER membrane is significantly more permeable than other cellular membranes and allows the passage of small charged molecules that are unable to cross the plasma membrane[41]. Therefore, BME-derived free biotin generated by CES1/2 in the ER quickly diffuses into the cytosol and is used by cytosolic or nuclear TurboIDs. Thus, the extremely high activity of TurboID, combined with the rapid substrate delivery of BME, provides researchers with a powerful method for capturing changes in protein interactions during rapid cell responses after drug administration or rapid changes in cellular conditions.

Biotin is a coenzyme of several carboxylases involved in various metabolic reactions and plays a role in the regulation of gene expression, cell proliferation, and survival[42]. The uptake of biotin, pantothenic acid, and lipoic acid is mediated by SMVT[15,17]. Biallelic mutations in *SMVT* are responsible for a recently described multivitamin-responsive inherited metabolic disorder, whose phenotypic spectrum and response to treatment

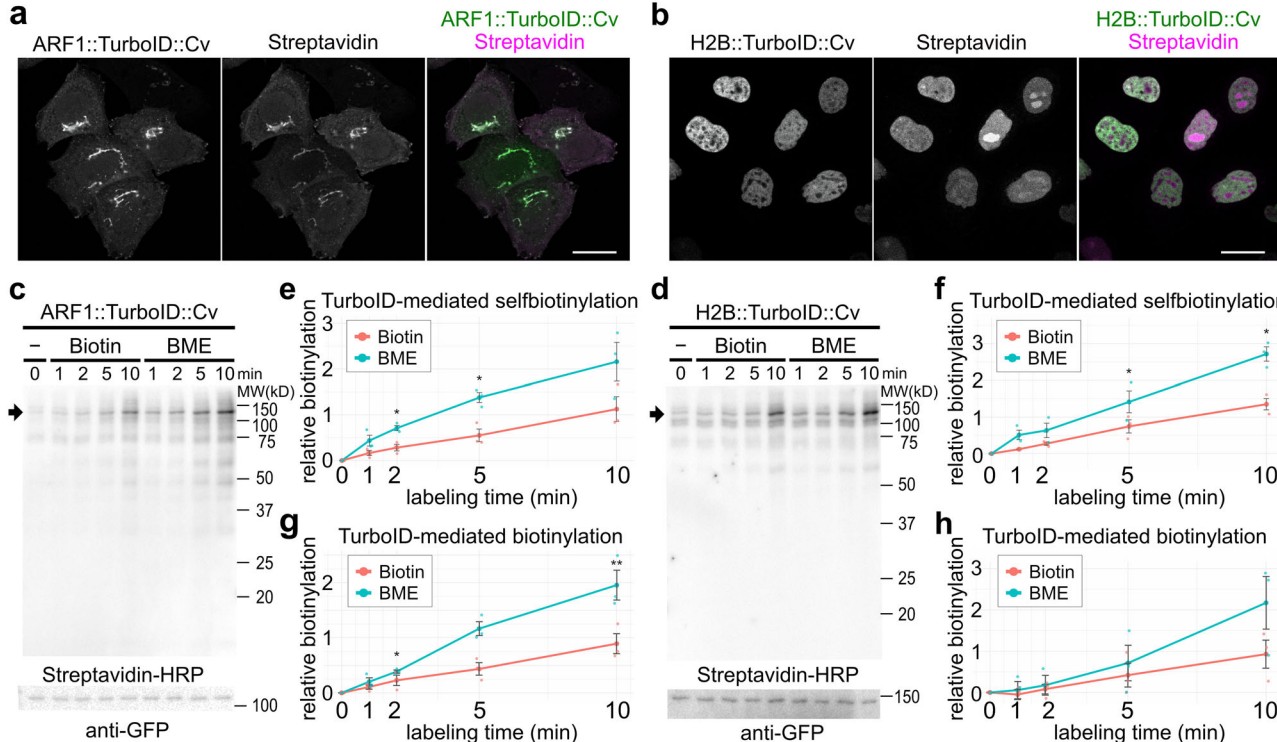

**Fig. 4 | BME enables the rapid biotinylation by cytosolic and nuclear TurboID in HeLa cells. a** Localization of ARF1::TurboID::Cv (green) and the biotinylated proteins (magenta). **b** Localization of H2B::TurboID::Cv (green) and the biotinylated proteins (magenta). Detection of biotinylated proteins by immunoblotting. Extracts from HeLa cells expressing ARF1::TurboID::Cv (**c**) or H2B::TurboID::Cv (**d**) were prepared after 0, 1, 2, 5, and 10 min incubation with biotin or BME at 50 µM. Streptavidin-HRP was used to detect biotinylated proteins (top), and the arrow indicates the band for ARF1::TurboID::Cv (**c**) or H2B::TurboID::Cv (**d**). An anti-GFP antibody was used to detect ARF1::TurboID::Cv (**c**) or H2B::TurboID::Cv (**d**) (bottom). Relative amounts of self-biotinylated ARF1::TurboID::Cv (**e**) and H2B::TurboID::Cv (**f**). The intensities of ARF1::TurboID::Cv and H2B::TurboID::Cv (red rectangles in Fig. S5d, e) were measured in three independent experiments. Significance was determined using a two-tailed paired Student's $t$ test. Exact $P$ values in **e** are 0.00814 (1 min), 0.0158 (2 min), 0.0137 (5 min), and 0.0740 (10 min). Exact

$P$ values in (**f**) are 0.1996 (1 min), 0.0816 (2 min), 0.0170 (5 min), and 0.0217 (10 min). **g, h** Relative amount of biotinylation by ARF1::TurboID::Cv (**g**) and H2B::TurboID::Cv (**h**). The intensity of the large areas on the immunoblot membrane (blue rectangles in Fig. S5d, e) were measured in three independent experiments. Significance was determined using a two-tailed paired Student's $t$ test. Exact $P$ values in (**g**) are 0.2606 (1 min), 0.181 (2 min), 0.0534 (5 min), and 0.0099 (10 min). Exact $P$ values in (**h**) are 0.683 (1 min), 0.747 (2 min), 0.607 (5 min), and 0.1819 (10 min). Because the signal intensity measurement for the 1-min incubation with biotin was below 0 (i.e., lower than the background level) in one of the H2B::TurboID::Cv experiments, we did not show the fold-change ratio (biotin versus BME) in the results. Error bars are presented in (**e–h**) as mean ± SE. Significance was determined using a two-tailed paired Student's $t$ test; *$p < 0.05$. Scale bars: 20 µm. BME, biotin methyl ester.

remain to be elucidated[43–46]. As BME readily penetrates cells in an SMVT-independent manner and is rapidly hydrolyzed to biotin, it could be an effective biotin supplement for patients with inherited metabolic disorders. This is analogous to the manner in which other prodrug medicines work against diseases[47,48].

## Methods
### Construction of plasmids for organelle localizing TurboIDs
**CT7-HA-Tb-2Cv**. To construct **CT7-HA-LL-Tb-2Cv**, a DNA fragment encoding TurboID was amplified from the plasmid pcDNA3-mem-TurboID (a gift from Jonathan Long, Addgene plasmid # 149409) using the primers (GGTGCGGCCGCCAAAGACAATACTGTGCCTCTG AAGC) and (GGTGCGGCCGCCAAAGACAATACTGTGCCTCTG AAGC), digested with NotI and BglII, and then cloned into NotI/BglII-digested **CT7-HA-2Cv**.

**CT7-HA-LL-Tb-2Cv**. To construct CT7-HA-LL-Tb-2Cv, a 65-aa long linker (GGGGSGGGGAGSGSGSGGGGSGGGGSGGGSGGGSGGGGSGG GSGGGGSGGGGSGGGSGGGTGTGTGT) was amplified from pEBP-Sec61b-mTagBFP2-LL-LOV2[14] using the primers (CCACTAGTGGAG GAGGAGGTTCTGGTGGTGGTGCGGGTTCTGGAAGTGGATCTG GAGGT) and (GGGGCGCGCCACCACCAGATCTACCACCAGCGG

CCGCTGTTCCTGTTCCGGTCCCCGT), digested with SpeI and NotI and cloned into the SpeI/NotI site of **CT7-HA-Tb-2Cv**.

**CT7-ARF1-LL-Tb-2Cv**. CT7-ARF1-LL-Tb-2Cv is a mammalian vector that expresses ARF1::TurboID::Cv. To construct this vector, the coding region of human *ARF1* was amplified from HeLa cDNA using the primers (GGGGTACCATGGGGAACATCTTCGCCAAC and CCAC-TAGTACCCTTCTGGTTCCGGAGCTGATT), digested with KpnI and SpeI, and cloned into CT7-HA-LL-Tb-2Cv.

**CT7-SP-2Cv-KDEL**. CT7-SP-2Cv-KDEL is a mammalian vector that expresses Cv::TurboID::KDEL in the ER lumen. To construct this vector based on CMV-SP-SBP-EGFP-GPI[14], a signal peptide of human IL2 from STR-KDEL_SBP-EGFP-GPI (gift from Franck Perez, Addgene plasmid # 65294), two copies of mClover3 from pKanCMV-mClover3-mRuby3 (gift from Michael Lin, Addgene plasmid # 74252), a 95-aa long linker from pEBP-Sec61b-mTagBFP2-LL-LOV2[14], and synthetic oligonucleotides coding "NIAAKDEL-STOP" were assembled.

**CT7-H2B-LL-Tb-2Cv**. CT7-H2B-LL-Tb-2Cv is a mammalian vector that expresses H2B::TurboID::Cv. To construct this vector, the human Histone-2B gene was amplified from plasmid pcDNA5/FRT/

**Fig. 5 | The models for improvements in the RUSH system and TurboID by BME. a** A model for improving the RUSH system using BME. In the RUSH system, the SBP-fused cargo is trapped in the ER, as SBP binds to streptavidin via an ER retention signal, KDEL. Administration of biotin or BME induces the dissociation of SBP from streptavidin, allowing the cargo to exit the ER simultaneously. Due to its high permeability, BME quickly occupies the SBP-binding sites on streptavidin. Conversely, biotin is taken up by SMVT, but it takes time to fill the site. **b** A model for improved biotinylation of TurboID using BME. TurboID is a promiscuous biotinylation enzyme with a high activity. TurboID produces the highly reactive biotinyl-5'-AMP from ATP and biotin. BME quickly permeates the plasma and ER membranes and is hydrolyzed to biotin by carboxylesterases. As the ER membrane is permeable to small, charged molecules, biotin quickly diffuses into the cytosol. In contrast, SMVT slowly transports biotin from the extracellular space to the cytosol. This limits the availability of free biotin for TurboID, even in the cytosol.

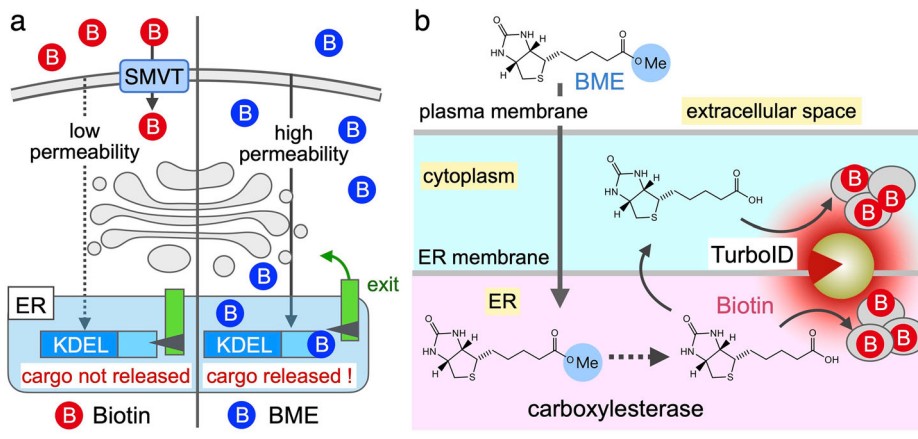

TO_H2B-HaloTag10-T2A-Tomm20-HaloTag9 (gift from Kai Johnsson, Addgene plasmid # 175525) using the primers H2B-Sp (CCAC-TAGTCTTAGCGCTGGTGTACTTGGTGAT) and K-CT7-H2B(CAC TATAGGGGATCCGGGGTACCATGCCAGAGCCAGCGAAGTC), digested with *Spe*I and *Kpn*I, and cloned into CT7-ARF1-LL-Tb-2Cv.

**CT7-hCES2.** CT7-hCES2 is a mammalian vector expressing human CES2. To construct this vector, the coding region of human *ARF1* was amplified from HeLa cDNA using the primers, hCES2-GF1 (AGCGAA CCGAGACCAGCGAG) and hCES2-GR1 (TCTCCTTAGTGGGTGT GTGTGGG), then amplified with K-CT7-hCES2 (CACTATAGGGG ATCCGGGGTACCACCATGCGGCTGCACAGAC) and hCES2-CT7-Ap (TAGGCTTACCTTCGAAGGGCCCCTACAGCTCTGTGTGTCT CTCTT), and then cloned between *Kpn*I and *Apa*I sites of CT7-ARF1-LL-Tb-2Cv using Gibson assembly.

**Synthesis of BME**
Biotin 51.2 mg (0.2095 mmol, FUJIFILM Wako Pure Chemical Corporation, Osaka, Japan) was suspended in 1.2 mL of MeOH, to which 45 µL of thionyl chloride (0.6241 mmol, Nacalai Tesque, Kyoto, Japan) was added under a nitrogen atmosphere. The mixture was stirred overnight at room temperature, and the solvent and excess thionyl chloride were removed under reduced pressure. The resulting crude product was purified by silica gel column chromatography (FL100D: EtOAc/MeOH = 6:1, Fuji Silysia Chemical, Kasugai, Japan) to give 51.1 mg (0.1978 mmol) of BME (94%). $^1$H NMR (Varian 400MR, Agilent, Santa Clara, USA; 400 MHz, CDCl$_3$) δ 5.55 (1H, d, $J$ = 5.7 Hz), 5.15 (1H, d, $J$ = 4.0 Hz), 4.53–4.50 (1H, m), 4.33–4.30 (1H, m), 3.67 (3H, s), 3.18–3.14 (1H, m), 2.92 (1H, dd, $J$ = 12.8, 5.0 Hz), 2.74 (1H, d, $J$ = 12.8 Hz), 2.34 (2H, t, $J$ = 7.5 Hz), 1.75–1.62 (4H, m), 1.49–1.42 (2H, m). $^{13}$C NMR (Varian 400MR, Agilent; 100 MHz, CDCl$_3$) δ 174.2, 107.6, 62.1, 60.4, 55.5, 51.8, 40.8, 33.8, 28.5, 28.4, and 24.9. HRMS (LTQ Orbitrap XL mass spectrometer, Thermo Fisher Scientific, Waltham, MI, USA; ESI) m/z [M + Na]$^+$ was calculated. for [C$_{11}$H$_{18}$N$_2$O$_3$NaS]$^+$ 281.09303, found 281.09290. FTIR (JASCO FT/IR-6300 Fourier transform infrared spectrometer, JASCO, Tokyo, Japan; NaCl) $v$ = 3272 broad, 3195 broad, 3114 broad, 3061, 2945, 2923, 2881, 2850, 1745, and 1710 cm$^{-1}$.

**Live imaging of the cells using the RUSH system with an FV3000 confocal microscope**
We used HeLa cells stably expressing GalT::iRFP713[14], MDCK cells, and HEK293T cells. HeLa cells stably expressing GalT::iRFP713[14] were

transfected with a DNA plasmid encoding the RUSH system bicistronic expression plasmid Str-KDEL_SBP-EGFP-GPI (a gift from Franck Perez, Addgene plasmid # 65294) using the JetOptimus transfection reagent (Polyplus-transfection, Illkirch-Graffenstaden, France), according to the manufacturer's instructions. MDCK and HEK293T cells were cotransfected with a DNA plasmid encoding GalT::iRFP680. In the experiments for Figure S2, HeLa cells were cotransfected with a DNA plasmid encoding Ruby::GM130[14]. At 6–10 h after transfection, the medium was replaced with fresh medium containing 25 µM of biliverdin (Cayman Chemical, Ann Arbor, MI, USA). The following day, SBP::EGFP::GPI was released into the secretory pathway by replacing the medium with BME (prepared in this study) or biotin (FUJIFILM Wako Pure Chemical Corporation).

A 1 mM concentration of Pantothenic acid (FUJIFILM Wako Pure Chemical Corporation) was added to the medium containing BME or biotin. Time-lapse confocal micrographs were obtained using an FV3000 confocal microscope (UPLXAPO60XO 1.30 NA and UPlanSApo 60 × S2 1.42 NA objective lens; Evident Scientific, Tokyo, Japan). To minimize bleed-through, SBP:: EGFP::GPI and GalT::iRFP680/713 or Ruby::GM130 signals were sequentially imaged. The images were processed in accordance with the Guidelines for Proper Digital Image Handling using ImageJ and/or Affinity Photo (Serif Europe Ltd., Nottingham, UK)[49].

**Live imaging of HeLa cells using the RUSH system with epi-fluorescent microscope**
HeLa cells (RIKEN BioResource Research Center, #RCB0007) were transfected with DNA plasmids encoding Str-KDEL_SBP-mCherry-GPI (gift from Franck Perez, Addgene plasmid # 65295) and EGFP-ST using the Lipofectamine 3000 (Thermo Fisher Scientific) transfection reagent according to the manufacturer's instructions. The plasmid encoding EGFP-ST was generated from mCherry-ST (a gift from Michael Davidson, Addgene plasmid # 55133) by replacing mCherry with EGFP using an In-Fusion Cloning kit (Takara Bio, Kusatsu, Japan). Following overnight incubation, epifluorescence micrographs were obtained using an IX83 microscope (Evident Scientific) equipped with a UPLXAPO100XO objective lens (1.45 NA, Evident Scientific) and a Zyla5.5 sCMOS camera (Andor Technology, Belfast, UK). Images were acquired at multi-point time-lapses (~20 stage points/dish, 2 min interval, and 61 time points) using the MetaMorph software (Molecular Devices, San Jose, USA). Immediately before time-lapse imaging, 20 µL of 100× biotin/BME solution (final concentration: 0.1–40 µM) was added to 2 ml of the culture medium. Biotin/BME was dissolved in PBS/DMSO to obtain a vehicle concentration of

<0.04%. A circular region of interest (ROI) was drawn on the Golgi ribbon area in the cell based on the EGFP-ST fluorescence image (green channel), and the time required to reach the maximum value of mCherry::GPI fluorescence (red channel) in the ROI after biotin/BME administration was determined.

### Detection of biotinylated proteins in cells via FV3000

HeLa cells stably expressing GalT::iRFP713 were transfected with a DNA plasmid encoding Cv::TurboID::KDEL using JetOptimus (Polyplus-transfection) transfection reagent according to the manufacturer's instructions. At 6–10 h after transfection, the medium was replaced with fresh medium containing 25 µM of biliverdin (Cayman Chemical). The following day, biotinylation was initiated using TurboID by replacing the medium with BME or biotin (FUJIFILM Wako Pure Chemical Corporation). Biotinylation was terminated by replacing the culture medium with 4% paraformaldehyde in PBS. The fixed cells were washed three times, permeabilized with 1% Triton in PBS for 2 min, and washed three times After blocking with 10% fetal bovine serum (FBS) and 0.04% sodium azide in PBS for 10 min, the cells were incubated with Alexa Fluor 568-conjugated streptavidin (1/500, Thermo Fisher Scientific). Fluorescent signals were imaged sequentially using an FV3000 confocal microscope (UPLXA-PO60XO 1.30 NA and UPlanSApo 60×S2 1.42 NA objective lens; Evident). The images were processed in accordance with the Guidelines for Proper Digital Image Handling using ImageJ and/or Affinity Photo (Serif Europe Ltd)[49].

To quantify biotinylation, aggregates were removed from the original image and the image resolution was reduced to 256 × 256 pixels using bilinear interpolation. Red (Streptavidin-Alexa568) and green (Cv::TurboID::KDEL) intensity values for each pixel were obtained from the processed image. Using the intensity values of each pixel, scatter plots were created in R (R Core Team, https://www.R-project.org/) using packages dplyr (https://dplyr.tidyverse.org) and ggplot2 (https://ggplot2.tidyverse.org). Pixels with green intensity values (Cv::TurboID::KDEL) greater than the background were used as data. The regression lines for the scatter plots were calculated using a simple linear regression model.

### Detection of biotinylated proteins via blotting

We used HeLa cells stably expressing GalT::iRFP713[14] and HEK293T cells. Cells were transfected with a DNA plasmid encoding Cv::TurboID::KDEL, ARF1::Cv::TurboID or H2B::Cv::TurboID using JetOptimus (Polyplus-transfection) transfection reagent according to the manufacturer's instructions. For the experiments in Fig. S4 e and f, HEK293T cells were cotransfected with a DNA plasmid encoding Cv::TurboID::KDEL and hCES2 using JetOptimus (Polyplus-transfection) transfection reagent. For the experiments in Fig. S4 a and b, MDCK cells were transfected with a DNA plasmid encoding Cv::TurboID::KDEL using JetPrime (Polyplus-transfection) transfection reagent. The next day, transfected cells were reseeded in 24-well plates. Twenty hours after reseeding, biotinylation was initiated using TurboID by replacing the medium with BME or biotin (FUJIFILM Wako Pure Chemical Corporation). To study the effects of BME and biotin concentrations, biotinylation was stopped by rapidly cooling the cells. The cooled cells were washed three times with ice-cold PBS and extracted by adding RIPA buffer (FUJIFILM Wako Pure Chemical Corporation) supplemented with benzonase nuclease (Sigma-Aldrich Corp., St. Louis, MO, USA) and protease inhibitor cocktail set V (FUJIFILM Wako Pure Chemical Corporation). After incubation for less than 2 min, biotinylation was stopped immediately by adding RIPA buffer containing benzonase nuclease and protease inhibitor cocktail set V after a quick wash with PBS.

Cell extracts or in vitro-biotinylated samples were separated by 10% acrylamide gel SDS-PAGE and blotted onto PVDF membranes (Millipore Sigma, Burlington, MA, USA). Cv::TurboID::KDEL or ARF1::Cv::TurboID were detected using anti-GFP (1/7500, REF: A6455, LOT: 1220284, Thermo Fisher Scientific), followed by HRP-conjugated anti-rabbit IgG (1/20000, Jackson ImmunoResearch Laboratories, West Grove, PA, USA).

HA::TurboID was detected using anti-HA (1/7500, REF: M180-3, LOT: 012, MBL life science, Tokyo, Japan), followed by HRP-conjugated anti-mouse IgG (1/20000, Jackson ImmunoResearch Laboratories).

Signals were visualized using enhanced chemiluminescence (Clarity Western ECL Substrate; Bio-Rad, Hercules, CA, USA) and imaged using ChemiDoc XRS+ (Bio-Rad). After the antibodies were stripped with WB Stripping Solution (Nacalai Tesque), the same PVDF membrane was incubated with HRP-conjugated streptavidin (1/20,000, Jackson ImmunoResearch Laboratories), and the signals were visualized. To measure self-biotinylation, the ROI was defined as shown in Fig. S3 and S5 (red squares) and the integrated density within the ROI was measured. To measure the biotinylation of other proteins, an ROI was defined as shown in Fig S3 (blue squares), and the integrated density within the ROI was measured. Biotinylation values were normalized to TurboID expression levels (detected using an anti-GFP antibody). The uncropped and unedited original blot images are presented in Figs. S6-11.

### In vitro biotinylation assay

DNA fragments encoding HA-TurboID with T7-promoter and SD-sequence were amplified from pCDNA3-mem-Turbo (Gift from Jonathan Long, Addgene plasmid number 149409), with primers "5'-GTTTAACTTTAAGAAGGAGATATACCAATGTA

CCCATACGATGTTCCAGATTACGCT-3", "5'-TAATACGACTCAC TATAGGGTTTT

GTTTAACTTTAAGAAGGAGATATACCAATG-3" and "5'-GAGCT AACGTGGCTT

CTTCTGCCAA-3"). mRNA transcription and protein synthesis were performed using PUREfrex 2.0 mini (GeneFrontier Corporation, Chiba, Japan) according to the manufacturer's instructions. Briefly, 90 µL of Transcription/Translation master mix containing 2 ng/µL of DNA template was incubated at 37 °C for 2 h. The mixture was aliquoted (4 µL) and placed on ice. To start biotinylation, BME (prepared in this study) or biotin (FUJIFILM Wako Pure Chemical Corporation) diluted in 1 µL of water was added and incubated at 37 °C for a designated time. Reaction was stopped by adding 5 µL of 2× SDS PAGE sample buffer.

### qPCR

Cell lysates were prepared using the SuperPrep II Cell Lysis & RT Kit for qPCR (TOYOBO, Osaka, Japan) according to the manufacturer's instructions. In brief, cultured human cells of about 2 cm² were washed twice with ice cold PBS and then lysed with 200 µL of lysis solution supplemented with gDNA remover and RNase inhibitor for 5 min at RT. The sample was centrifuged at 15000 rpm for 5 min to remove debris. Then, 20 µL of each lysate was added to 100 µL of RT master mix, and reverse transcription was performed to generate cDNA at 42 °C for 15 min and 50 °C for 10 min. Preparation of the lysate and RT were repeated three or more times for each cell line. Quantitative PCR was performed using the Dice Real-Time System III TP990 (Takara Bio) and the KOD SYBR qPCR Kit (TOYOBO). For each primer pair, the efficacy of amplification per cycle was confirmed by amplifying a serially diluted mixture of all samples, and the housekeeping gene beta-actin (*ACTB*) was chosen as the control. We applied a thermocycling program of 98 °C for 3 min, 98 °C for 10 s, 60 °C for 10 s, and 68 °C for 30 s, with 45 cycles of steps 2–4, with primers hACTB-qF1, 5′-CAC-CATTGGCAATGAGCGGTTC-3′, hACTB-qR1, 5′-AGGTCTTTGCG-GATGTCCACGT-3′, hSLC5A6-qF1, 5′-GGCTGCTTTGTCTAGGAAT GGC-3′, hSLC5A6-qR1, and 5′-CATTCCAAGGCAGAAGAGTCCC-3′. The fold change in the target gene expression over that of *ACTB* gene in HeLa cells was calculated using the standard dCt method.

### Statistics and reproducibility

R (4.3.1) using packages dplyr (https://dplyr.tidyverse.org) and ggplot2 (https://ggplot2.tidyverse.org), Prism9 software (GraphPad) was used to analyze the data. Data are shown as means ± SD or SE described in the legends of each graph. Statistical tests, statistical significance are described in the legends of each graph. All blotting experiments were performed at least

**Article**

three times. The signal intensities of images taken by confocal microscopy were measured for five images at each point.

## Reporting summary

Further information on research design is available in the Nature Portfolio Reporting Summary linked to this article.

## Data availability

Values for all data points found in graphs can be found in the 'Supplementary data' file. Additional information and other data are available from the corresponding author upon request.

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

## Acknowledgements

This work was supported by the Japan Society for the Promotion of Science (JSPS) (KAKENHI grant no. 22H02617) to A.K.S., (KAKENHI grant no. 19K06566) to T.S., (KAKENHI grant no. 21K05413) to T.N., (KAKENHI grant nos. 19H04764 and 22K06213) to T.T., and the Japan Science and Technology Agency (CREST grant no. JPMJCR22E2) to A.K.S., (CREST grant no. JPMJCR21E3) to T.T., (SPRING Grant No. JPMJSP2132) to A.T., and Core Research for Organelle Diseases funded by Hiroshima University, Takeda Science Foundation, and Ohsumi Frontier Science Foundation to A.K.S. We thank Dr. Yasuhiro Ishihara (Hiroshima University) for kindly providing cDNAs from several human cell lines. We also thank Editage (http://www.editage.com) for editing and reviewing this manuscript.

## Author contributions

T.S. and A.S. designed the study; U.T. performed the experiments and A.T. supervised the experiments and analyzed the data for TurboID; A.T. and T.T. performed the experiments and analyzed the data for the RUSH system; A.N. supervised the experiments for the RUSH system; S.K. performed the organic chemistry experiments; T.N. supervised the organic chemistry experiments; T.S. supervised the molecular biology experiments; and A.S. supervised all aspects of the project. T.S. and A.S. wrote the paper with input and final approval from U.T., A.T., T.T., A.N., S.K., and T.N.

## Competing interests

The authors declare no competing interests.
