## [Transparent Peer Review file · Communications Biology]

Biotin methyl ester enhances cargo release in RUSH system and enables rapid biotinylation with TurboID

Corresponding Author: Dr Akiko Satoh

Version 0:

Reviewer comments:

Reviewer #1

(Remarks to the Author)

Uehara et al. focused on the cellular application of biotin in the RUSH system and other biotinylation systems like TurboID. They reported that the biotin methyl ester (BME) with its masked carboxyl group can improve cell penetration, followed by enhanced cargo release in RUSH as well as faster biotinylation in TurboID. Given the wide applications of RUSH and TurboID, this finding is practical (quite easy to employ) and also very useful (with > 2-fold improvement compared to the acidic biotin at different concentrations) for researchers in this area. Meanwhile, the authors have carried out a systematic analysis to illustrate how BME achieved such improvement. Most of the conclusions have been well supported by the evidence and the manuscript is well presented. Following are several suggestions.

1. The author mentioned that "biotin has a carboxyl group with an acid dissociation constant of 4.7, and therefore, carries a positive charge at physiological pH". The biotin with a carboxyl acid should be in the COO⁻ form (with a negative charge) but not with a positive charge at physiological pH?
2. Is there a more direct method to confirm the enhanced penetration/cellular uptake of BME in comparison to biotin? For example, the cell penetration assay, or LCMS analysis of cell lysates? Related references for BME can also be added.
3. Although the expression of CES1 in HeLa cells was much lower than that in other cells, it is suggested to show the labeling effect of TurboID in other cells with BME in comparison with biotin, to support the relationship between CES1 and the labeling effect.
4. The authors mentioned that "To investigate this possibility, we examined the effects of BME on the rapid biotinylation by TurboID localized in the cytoplasm and nucleus", then ARF1::TurboID::Cv and Histone H2B(H2B)::TurboID::Cv were constructed. While ARF1::TurboID::Cv was localized mainly in the Golgi apparatus and slightly in the cytoplasm. Since this localization was "as expected", the reviewer thinks it may be better to indicate at the beginning "to examine the effects of BME on the biotinylation by TurboID localized in the Golgi and nucleus"? As long as not in the ER, Golgi or cytoplasm should both be fine? The description may be optimized.
5. The authors mentioned that "Possible BME-hydrolyzing enzymes, CES1 and CES2, are localized in the ER, resulting in a higher concentration of biotin in the ER lumen than in the cytoplasm or other organelles. Nevertheless, our comparison of the effects of BME to those of biotin on the biotinylation activity of TurboID localized in the ER, cytoplasm, and nucleus indicated that wherever TurboID is localized, BME strongly enhances biotinylation rates." It was indicated that the improvement in the RUSH system and TurboID biotinylation using BME instead of biotin is due to the improved cell penetration of BME. Meanwhile, the hydrolyzed BME (in the form of acidic biotin) seems to diffuse well inside cells, i.e., from ER to other subcellular locations. Can the authors give more discussions/explanations on the penetration of biotin into cells vs. the diffusion inside cells between other membrane-encapsulated organelles?
6. For cell-based experiments, it may not be suitable to use "in vivo", which is often used for animal-based experiments.

Reviewer #2

(Remarks to the Author)

The authors present data showing biotin methyl ester (BME) is a better reagent for stimulating cargo release in the RUSH system than biotin and similarly because of its rapid cellular uptake, a better substrate for biotin ligase. As pointed out by the authors in the discussion these results would seem to be important methodological developments for those working with the RUSH or BirA systems.

Overall the manuscript is clear even to a non-expert in the microscopy/cellular biology field like myself. The authors describe the findings of their experiments well and seem to have done many control experiments to verify both the major claims in the paper, i.e. BME is a better substrate for both RUSH and BirA like enzymes in vivo.

I like that the authors don't just describe the observation but also have done some work to suggest a mechanism for the difference in Biotin and BME's efficiency. Passive vs active cellular uptake and hydrolysis of the ester by CES1/2

I am little unsure about the practicality of using of BME as a therapeutic for metabolic disease, a reference showing it can be administered safely to mammals or some precedence for its use therapeutically would help bolster this statement.

I think the manuscript (length permitting) would benefit from some kind of high level simplified Figure that shows the principles of the RUSH and BirA systems and how BME improves the experiment.

Reviewer #3

(Remarks to the Author)

The manuscript by Uehara et al. evaluates the performance of biotin methyl ester (BME) and biotin in cargo release within the RUSH system, as well as proximity protein biotinylation using TurboID. The results indicate that BME penetrates the cell membrane more rapidly than biotin, initiating cargo release within 20 minutes and thus enabling more synchronized cargo release across cells. The authors further demonstrate that BME is hydrolyzed into biotin by intracellular carboxylesterases, allowing it to function as a biotin prodrug that facilitates faster biotinylation by TurboID.

Overall, the manuscript is well-written, and the conclusions are generally supported by the data. However, several issues should be addressed:

Major issues:

1. The application of BME in the RUSH system has been previously reported (JACS Au 2021, 1, 2, 221–232; J. Am. Chem. Soc. 2020, 142, 10, 4784–4792). The authors should reference these studies in the introduction and clarify the novelty and significance of their current work.
2. The manuscript highlights the rapid cellular uptake of BME as a key advantage over biotin. However, it is unclear whether this is solely due to BME's reduced hydrophilicity, leading to increased membrane permeability, or whether its uptake is still mediated by the biotin transporter SMVT. The authors should address this point explicitly.
3. Although SLC5A6 and CES1/2 expression levels were compared across different cell lines, the functional evaluation of BME and biotin was limited to HeLa cells. To strengthen their claims, the authors should include data from additional cell lines, at least with microscopic characterization, to confirm that BME consistently outperforms biotin.
4. The criteria for selecting bands representing non-autobiotinylation appear arbitrary. The authors should justify the choice of the blue-highlighted region and explain how it reliably reflects non-autobiotinylated proteins.

Minor issues:

1. It can be inferred that HeLa cells were used in Figure 1, but this should be explicitly stated in the figure legend or main text.
2. Figures 1b and 1c present data from different probes (SBP::GFP::GPI and SBP::NG::GPI, respectively). To ensure consistency, both image and quantification should be based on the same probe.
3. If interpreted correctly, cells that had not reached maximum intensity by 120 minutes were counted as ">120 min." This should be clarified, as it seems counterintuitive.
4. Since BME is commercially available, the rationale for synthesizing it in-house should be explained.

Version 1:

Reviewer comments:

Reviewer #1

(Remarks to the Author)

The authors have adequately addressed the reviewers' concerns.

Reviewer #2

(Remarks to the Author)

Thank you to the authors for the addition of a clarifying figure 5 to demonstrate the principle of improved BME performance.

Apart from a few minor suggestions I think the manuscript is suitable for publication. It seems to be a major methodological development that users of the RUSH system should be aware of.

Suggested alterations

1. Page 7 line 148

"Although BME is expected to increase membrane permeability, its uptake may be mediated by SMVT"

This statement sounds like BME increases the general membrane permeability of the cell, I think something like the below is clearer

"Although the BME molecule is expected to have increased membrane permeability compared to biotin, its uptake may still be mediated by SMVT"

2. PAGE 14 line 330

"However, this was unexpected following biotin administration in the RUSH system."

I found this sentence a little confusing. To clarify, I think the authors mean that the starting point of biotin transport initiation is much more varied across a population of fixed cells when using biotin rather than BME in the RUSH system?

Changing the sentence to something like the below might be helpful

"Whereas with biotin administration, the timing of transport initiation in different fixed cells is much more varied in the RUSH system."

Reviewer #3

(Remarks to the Author)

The revised manuscript has addressed the concerns.

Reviewer #1 (Remarks to the Author):

Uehara et al. focused on the cellular application of biotin in the RUSH system and other biotinylation systems like TurboID. They reported that the biotin methyl ester (BME) with its masked carboxyl group can improve cell penetration, followed by enhanced cargo release in RUSH as well as faster biotinylation in TurboID. Given the wide applications of RUSH and TurboID, this finding is practical (quite easy to employ) and also very useful (with > 2-fold improvement compared to the acidic biotin at different concentrations) for researchers in this area. Meanwhile, the authors have carried out a systematic analysis to illustrate how BME achieved such improvement. Most of the conclusions have been well supported by the evidence and the manuscript is well presented. Following are several suggestions.

1. The author mentioned that “biotin has a carboxyl group with an acid dissociation constant of 4.7, and therefore, carries a positive charge at physiological pH”. The biotin with a carboxyl acid should be in the COO⁻ form (with a negative charge) but not with a positive charge at physiological pH?

Thank you for pointing that out. It's an embarrassing mistake. I have corrected the text.

2. Is there a more direct method to confirm the enhanced penetration/cellular uptake of BME in comparison to biotin? For example, the cell penetration assay, or LCMS analysis of cell lysates? Related references for BME can also be added.

Thank you for your advice. We sought the way to quantify intracellular concentration of BME, however, we think that there is no practical assay to perform.

3. Although the expression of CES1 in HeLa cells was much lower than that in other cells, it is suggested to show the labeling effect of TurboID in other cells with BME in comparison with biotin, to support the relationship between CES1 and the labeling effect.

In this revised manuscript, we investigated the effects of BME using HEK293T and MDCK cells, in addition to HeLa cells. We demonstrated that BME strongly accelerates biotinylation compared to biotin in MDCK cells. However, it is difficult to compare CES1 and CES2 expression levels between canine cells and human cells (MDCK and HeLa cells, respectively). Conversely, we observed limited acceleration effects of BME in HEK293T cells; the accelerated biotinylation by BME was only seen on the very short incubation time points (1- and 2-min). Although qPCR indicates higher CES1 and CES2 expression levels in HEK293T cells than in HeLa cells, others have reported low CES1 expression levels and carboxylesterase activity in HEK293T cells. Thus, we examined whether CES2 overexpression strengthens the BME effects on biotinylation. We detected the accelerated biotinylation by BME at longer incubation time points in HEK293T cells expressing CES2 than in HEK293T cells not expressing CES2. These results support the relationship between CES2 and the labeling effect. We added the new section in results: its title

is "BME enables rapid TurboID biotinylation in HEK293T and MDCK cells".

4. The authors mentioned that "To investigate this possibility, we examined the effects of BME on the rapid biotinylation by TurboID localized in the cytoplasm and nucleus", then ARF1::TurboID::Cv and Histone H2B(H2B)::TurboID::Cv were constructed. While ARF1::TurboID::Cv was localized mainly in the Golgi apparatus and slightly in the cytoplasm. Since this localization was "as expected", the reviewer thinks it may be better to indicate at the beginning "to examine the effects of BME on the biotinylation by TurboID localized in the Golgi and nucleus"? As long as not in the ER, Golgi or cytoplasm should both be fine? The description may be optimized.

We prefer to state "the cytoplasm/cytosol" rather than "the Golgi" for the localization of ARF1 in the beginning, because readers might imagine the lumen of the Golgi if we state "the Golgi." Since vesicle transport connects the Golgi lumen to the ER lumen, where CES1 and CES2 are localized, we would like to emphasize that the protein is localized on the cytoplasmic face of the Golgi apparatus. However, we think "cytosol" is better expression than "cytoplasm" to describe the ARF1 localization and we changed "cytoplasm" to "cytosol" in the text.

5. The authors mentioned that "Possible BME-hydrolyzing enzymes, CES1 and CES2, are localized in the ER, resulting in a higher concentration of biotin in the ER lumen than in the cytoplasm or other organelles. Nevertheless, our comparison of the effects of BME to those of biotin on the biotinylation activity of TurboID localized in the ER, cytoplasm, and nucleus indicated that wherever TurboID is localized, BME strongly enhances biotinylation rates." It was indicated that the improvement in the RUSH system and TurboID biotinylation using BME instead of biotin is due to the improved cell penetration of BME. Meanwhile, the hydrolyzed BME (in the form of acidic biotin) seems to diffuse well inside cells, i.e., from ER to other subcellular locations. Can the authors give more discussions/explanations on the penetration of biotin into cells vs. the diffusion inside cells between other membrane-encapsulated organelles?

Thank you for your advice. We add the following discussion.

"The ER membrane is significantly more permeable than are other cellular membranes and allows the passage of small charged molecules that are unable to cross the plasma membrane (PMID: 14617815). Therefore, BME-derived free biotin generated by CES1/2 in the ER quickly diffuses into the cytoplasm and is used by cytoplasmic or nuclear TurboIDs."

6. For cell-based experiments, it may not be suitable to use "in vivo", which is often used for animal-based experiments.

We changed "in vivo" to "in living cells".

Reviewer #2 (Remarks to the Author):

The authors present data showing biotin methyl ester (BME) is a better reagent for stimulating cargo release in the RUSH system than biotin and similarly because of its rapid cellular uptake, a better substrate for biotin ligase. As pointed out by the authors in the discussion these results would seem to be important methodological developments for those working with the RUSH or BirA systems.

Overall the manuscript is clear even to a non-expert in the microscopy/cellular biology field like myself. The authors describe the findings of their experiments well and seem to have done many control experiments to verify both the major claims in the paper, i.e. BME is a better substrate for both RUSH and BirA like enzymes in vivo.

I like that the authors don't just describe the observation but also have done some work to suggest a mechanism for the difference in Biotin and BME's efficiency. Passive vs active cellular uptake and hydrolysis of the ester by CES1/2

I am little unsure about the practicality of using of BME as a therapeutic for metabolic disease, a reference showing it can be administered safely to mammals or some precedence for its use therapeutically would help bolster this statement.

Unfortunately, we could not find previous studies investigating the safety or therapeutic use of BME in mammals. However, there is extensive literature on esterified prodrugs. Thus, we cited reviews of esterified prodrugs. We added the following sentences to the discussion:

" This is analogous to the manner in which other prodrug medicines work against diseases" (PMID: 40525803, 39940757).

I think the manuscript (length permitting) would benefit from some kind of high level simplified Figure that shows the principles of the RUSH and BirA systems and how BME improves the experiment.

Thank you for the suggestion. We have included the schematics in Figure 5 in our revised manuscript. This figure illustrates the principles of the RUSH system and TurboID, as well as how BME improves those systems.

Reviewer #3 (Remarks to the Author):

The manuscript by Uehara et al. evaluates the performance of biotin methyl ester (BME) and biotin in cargo release within the RUSH system, as well as proximity protein biotinylation using TurboID. The results indicate that BME penetrates the cell membrane more rapidly than biotin, initiating cargo release within 20 minutes and thus enabling more synchronized cargo release across cells. The authors further demonstrate that BME is hydrolyzed into biotin by intracellular carboxylesterases, allowing it to function as a biotin prodrug that facilitates faster biotinylation by TurboID.

Overall, the manuscript is well-written, and the conclusions are generally supported by the data. However, several issues should be addressed:

Major issues:

1. The application of BME in the RUSH system has been previously reported (JACS Au 2021, 1, 2, 221–232; J. Am. Chem. Soc. 2020, 142, 10, 4784–4792). The authors should reference these studies in the introduction and clarify the novelty and significance of their current work.

Thank you for letting us know those reports. We cited them and added the following sentences in the introduction.

“Of note, it has already been demonstrated that BME can release cargo in the RUSH system, although whether BME is more effective than biotin in cargo release has not been addressed.”

2. The manuscript highlights the rapid cellular uptake of BME as a key advantage over biotin. However, it is unclear whether this is solely due to BME's reduced hydrophilicity, leading to increased membrane permeability, or whether its uptake is still mediated by the biotin transporter SMVT. The authors should address this point explicitly.

To investigate whether BME is transported by the sodium multivitamin transporter SMVT, we have used pantothenic acid, an inhibitor for SMVT. Pantothenic acid inhibited the start of cargo release by biotin, but not the one by BME. Thus, the uptake of BME is not mediated by SMVT. We included this observation in Figure S2 and added the following sentences in the text.

“Although BME is expected to increase membrane permeability, its uptake may be mediated by SMVT. Thus, we investigated whether pantothenic acid, which is known to competitively inhibit biotin transport mediated by SMVT, would also inhibit the cargo release induced by BME. As expected, simultaneously adding 1 mM pantothenic acid severely inhibited cargo release induced by 40 μ M biotin (Supplementary Fig. S2b). In contrast, 40 μ M BME effectively released cargo in all cells, even in the presence of 1 mM pantothenic acid (Supplementary Fig. S2d). These results suggest that BME entry into cells does not depend on SMVT and that BME penetrates directly into cells due to its hydrophobicity.”

3. Although SLC5A6 and CES1/2 expression levels were compared across different cell lines, the functional evaluation of BME and biotin was limited to HeLa cells. To strengthen their claims, the authors should include data from additional cell lines, at least with microscopic characterization, to confirm that BME consistently outperforms biotin.

We performed RUSH assay and biotinylation by TurboID using biotin or BME in MDCK cells and HEK293T cells. BME consistently outperforms biotin for RUSH system and biotinylation in the short incubation time. However, the longer incubation than 5min with biotin showed higher biotinylation than that with BME in HEK293T cells. We also showed CES2 overexpression

improves biotinylation by BME in the long incubation time in HEK293T cells, suggesting that hydrolysis of BME could be the rate-limiting step for biotinylation in certain cell types. We included these data and discussion.

4. The criteria for selecting bands representing non-autobiotinylation appear arbitrary. The authors should justify the choice of the blue-highlighted region and explain how it reliably reflects non-autobiotinylated proteins.

Some bands were labeled independently of biotin/BME administration. Additionally, it is difficult to quantify bands with a weak signal because a relatively high background signal affects the real signal. For these reasons, we selected a region with higher molecular weight than 150 kD. In revised manuscript, we re-selected a region more carefully and re-measured the signal intensities.

Minor issues:

1. It can be inferred that HeLa cells were used in Figure 1, but this should be explicitly stated in the figure legend or main text.

We included the information of cell lines.

2. Figures 1b and 1c present data from different probes (SBP::GFP::GPI and SBP::NG::GPI, respectively). To ensure consistency, both image and quantification should be based on the same probe.

We revised the results in Figures 1c and 1d. We collected new data using "SBP::GFP::GPI" instead of "SBP::NG::GPI." However, after conducting these experiments and new experiments for Figures S1 and S2, we realized that we had used "SBP::mCherry::GPI" instead of "SBP::GFP::GPI" in Figure 1b. Our FV3000 is not very compatible with mCherry. Thus, we hope the reviewer will accept the different probes.

3. If interpreted correctly, cells that had not reached maximum intensity by 120 minutes were counted as ">120 min." This should be clarified, as it seems counterintuitive.

We were sorry that was the mistake. We changed Figure 1b legend "<120 min" from to ">120 min".

4. Since BME is commercially available, the rationale for synthesizing it in-house should be explained.

In fact, during the couple of years that we worked on this project, it was not easily available in Japan. Wako and other company used to sell BME but stopped selling it for several years in Japan. Now, they are selling it again. Another reason is that BME synthesis is quite easy and inexpensive.

Reviewer #1 (Remarks to the Author):

The authors have adequately addressed the reviewers' concerns.

Reviewer #2 (Remarks to the Author):

Thank you to the authors for the addition of a clarifying figure 5 to demonstrate the principle of improved BME performance.

Apart from a few minor suggestions I think the manuscript is suitable for publication. It seems to be a major methodological development that users of the RUSH system should be aware of.

Suggested alterations

1. Page 7 line 148

"Although BME is expected to increase membrane permeability, its uptake may be mediated by SMVT"

This statement sounds like BME increases the general membrane permeability of the cell, I think something like the below is clearer

"Although the BME molecule is expected to have increased membrane permeability compared to biotin, its uptake may still be mediated by SMVT"

Thank you for pointing that out. We agree with you. We have changed the text.

2. PAGE 14 line 330

"However, this was unexpected following biotin administration in the RUSH system."

I found this sentence a little confusing. To clarify, I think the authors mean that the starting point of biotin transport initiation is much more varied across a population of fixed cells when using biotin rather than BME in the RUSH system?

Changing the sentence to something like the below might be helpful

"Whereas with biotin administration, the timing of transport initiation in different fixed cells is much more varied in the RUSH system."

We have changed the text.

Reviewer #3 (Remarks to the Author):

The revised manuscript has addressed the concerns.